# Measurement of Tooth Wear by Means of Digital Impressions: An In-Vitro Evaluation of Three Intraoral Scanning Systems

Christina Kühne [1,*] , Ulrich Lohbauer [2], Stefan Raith [3] and Sven Reich [1]

1   Department of Prosthodontics and Biomaterials, RWTH Aachen University Hospital, Pauwelsstrasse 30, 52074 Aachen, Germany; sreich@ukaachen.de
2   Research Laboratory for Dental Biomaterials, Dental Clinic 1, University Hospital Erlangen, 91054 Erlangen, Germany; ulrich.lohbauer@fau.de
3   Department of Oral and Maxillofacial Surgery, RWTH Aachen University Hospital, Pauwelsstrasse 30, 52074 Aachen, Germany; sraith@ukaachen.de
*   Correspondence: chkuehne@ukaachen.de; Tel.: +49-2418085842

**Abstract:** This in-vitro study aimed to investigate whether intraoral scanners (IOS) are suitable for wear measurement compared to optical profilometry (WLP). A zirconia cast representing the teeth (24–28) was fabricated. It was digitized six times using three different intraoral scanners, Cerec Omnicam AC (OC), Trios 3 (Tr3), and True Definition (TD). The scans were conducted at baseline (t0) and at three different stages of simulated wear (t1–t3), each at one wear-facet on FDI 26 and FDI 27. WLP was used as a reference method. Within each acquisition system, the maximum wear at each facet was analyzed by superimposing the STL data of t0 with t1–t3. A power analysis was performed (G*Power), and the Wilcoxon-signed-rank-test was used to evaluate whether there were statistically significant differences between the groups (Bonferroni corrected) ($\alpha = 0.05$). At wear-facet FDI 27, differences from +4% t1 TD up to +19% t2 OC, corresponding to a metric value of 8 μm and 45 μm, were measured. At FDI 26 deviations between −2% t1 Tr3, and +10% OC and Tr3, were observed. Considering some limitations, the IOS are a promising alternative to wear measurement based on WLP due to its simple application to capture surface changes in a reasonable and quick way.

**Keywords:** digital dentistry; tooth wear; intraoral scanner; diagnostic imaging





## 1. Introduction

The evaluation and monitoring of tooth wear are essential for therapeutic strategies and scientific purposes. For scientific investigations, wear measurement is still predominantly based on a conventional workflow [1,2] by taking impressions and fabricating casts. Depending on the analytical method, different subsequent steps are essential. The cast can be digitized with a laboratory or an industrial high-resolution scanner [3–6]. The obtained 3D data files of different wear statuses are superimposed with 3D analyzing software to calculate metrical wear values. If optical profilometry is applied, the wear facets have to be identified by scanning electron microscopy (SEM). Then, the identified areas are captured by profilometry. The digital data files are superimposed, and vertical height loss is analyzed between two different stages of wear. In summary, conventional workflows are time-consuming and, therefore, far away from real-time analysis. Although single steps of the conventional workflow like SEM and profilometry are highly accurate, some other essential procedures like conventional impression taking and producing the casts limit the overall accuracy. Intraoral scanning offers direct access to 3D data files without the necessity of intermediate steps. In literature, the first insights towards implementing optical impression systems are reported [7,8]. Hartkamp et al. showed that in vitro wear analysis based on data files obtained with an intraoral scanner was comparable to optical profilometry. The Lava C.O.S.'s (Chairside Oral Scanner, 3M ESPE) optical principle was based on wavefront sampling, and "dusting" of the scanned surface with titanium dioxide

was required. Therefore, the study aimed to test if wear measurements based on data files obtained with powder-free scanners are comparable to results based on profilometry.

The null hypothesis stated was:

The exclusively digital workflow with data acquired on the basis of intraoral optical impression systems is equal to the white light profilometer for wear measurement within deviations of a maximum of 10%.

## 2. Materials and Methods

### 2.1. Digitization of the Study Cast

A zirconia (VITA YZ HT, VITA Zahnfabrik, Bad Säckingen, Germany) cast representing the posterior teeth of the left maxillary quadrant (FDI 24–28) (Figure 1) was produced.

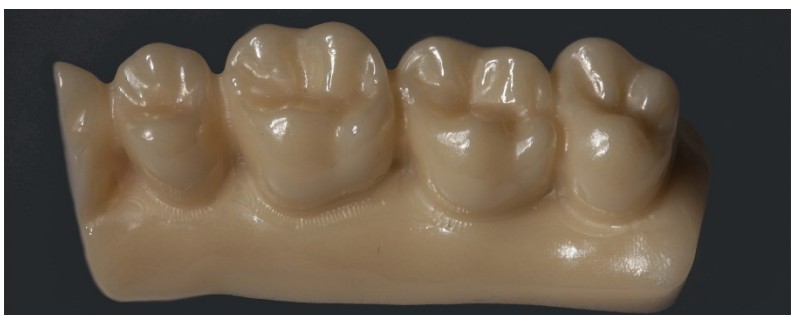

**Figure 1.** The zirconia model, consisting of the teeth 24–28, is shown at baseline.

It was based on the scan data of a typodont (Basic model, KaVo Dental GmbH, Biberbach, Germany). After milling, sintering, and polishing, the cast was scanned at baseline t0 using three intraoral scanners (IOS): Cerec Omnicam AC (OC) (V4.5, Dentsply Sirona Deutschland GmbH, Bensheim, Germany), Trios 3 (Tr3) (V.1.16.1.0, 3Shape A/S, Copenhagen, Denmark) and True Definition Scanner (TD) (S1548041, 3M Deutschland GmbH, Seefeld, Germany). The use of TD required a slight dusting with titanium dioxide particles (3M High-Resolution Scanning Spray, 3M Espe Dental products St. Paul, USA) in order to create randomly distributed landmarks for the optical system applying wavefront sampling [9]. In the case of the powder-free IOS, Tr3, and OC, scans of the powdered (p) and unpowdered (non-p) cast were accomplished in order to evaluate whether the modification of the surface influenced the data. Tr3 utilizes the imaging principle of confocal microscopy and OC uses active triangulation. To obtain a highly accurate digital reference, a high-resolution non-contact white light profilometer (WLP) (CT 100 and the sensor P-CHR-6000, z-res 200 nm Cybertechnologies, Ingolstadt, Germany) was used. The sensor of CT 100 provided a vertical resolution up to 3 nm and a lateral resolution of 0.05 μm [10]. Before capturing data with WLP, the model was air-abraded with 50 μm aluminum oxide at 2 bar. With both acquisition systems, IOS and WLP, the partial arch was captured. The sensor's spot size was 16 μm, combined with a 150 × 150 × 40 mm scanning area at maximum x, y, z resolution. The zirconia cast was always orientated on the WLP scanning platform in the identical occluso-apical direction (base of the zirconia cast). Simultaneously, the WLP could always capture the entire occlusal surface from the cusps to the anatomical equator. The IOS had a field of view of approximately 10 × 13 mm at a vertical working distance of 17 mm using TD, as an example. According to the manufacturers' recommendations, a complete partial-arch scan was conducted applying the IOS.

After all scans at t0 were accomplished, wear areas were manually simulated using a diamond-bur (Ø1.2 mm, grit size 46 μm) [11]. In total, three stages of wear (t1–t3) were created successively at tooth 26 mesio–oral and tooth 27 disto–buccal (Figure 2). At t1, t2, and t3, the cast was scanned again with the scanners mentioned above, and WLP. At each stage, the cast was scanned six times one after another by the same experienced operator with each intraoral scanning system. Before each scanning session, the scanning system

was calibrated according to the manufacturer's recommendations. Optical profilometry (WLP) served as a reference and was carried out once per stage.

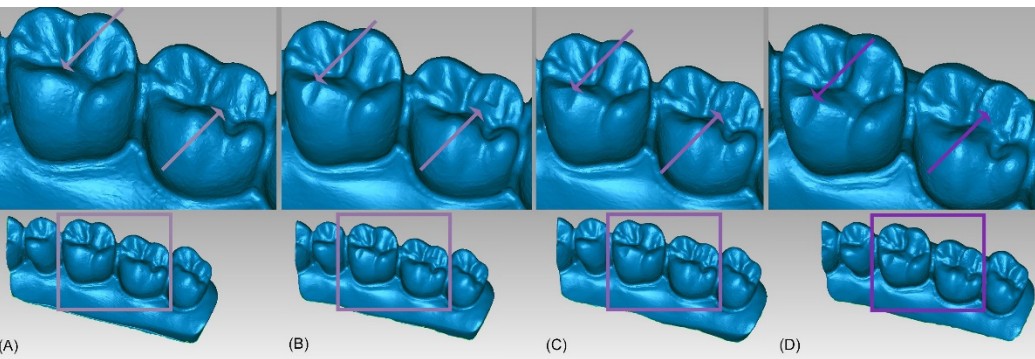

**Figure 2.** The study model was digitized and imported into Geomagic Qualify. The several wear steps are presented: Baseline t0 (**A**) and the wear stages [t1] (**B**), t2 (**C**), and t3 (**D**).

### 2.2. Analysis of the Obtained Data Using a Reference Best-Fit-Alignment

The captured 3D data were analyzed with Geomagic Qualify 2012 (V.2012_08_08_E, 64-bit, Morrisville, NC, USA). The files were imported as STL (Standard triangulation language) datasets. The surface of the obtained datasets consisted of a triangle mesh. In this mesh, one triangle adjoined other triangles, creating a vertex at each meeting point.

First, each dataset (t1–t3) was superimposed with its respective baseline data file. Before superimposition, the worn areas were virtually excluded in the software to achieve a more accurate best fit alignment (tolerance value 0.01 mm [8]) [12–14].

Thus as an example, OC p $t0/_{n(1-6)}$ was defined as a reference and registered with OC p $t1/_{n(1-6)}$, $t2/_{n(1-6)}$, and $t3/_{n(1-6)}$. The superposition of the datasets required the definition of a reference and a test dataset. The reference t0 was fixed in the coordinate system. Therefore, the test dataset t1–t3 could be positioned as identically as possible. The WLP was the key dataset, which determined the surface area to be superimposed since it has a smaller field of view than the IOS. To ensure comparability, the same area size was therefore included in the registration.

Figure 3A shows the result of the superimposition of two meshes according to best fit alignment. The bluish color-coded area indicates the wear facet between the baseline t0 and a stage of wear. The schematic drawing in Figure 3B displays the distance measurement principle between two datasets: a perpendicular was dropped from one vertex of the baseline dataset to a corresponding triangular facet that belongs to another dataset where a wear facet was simulated.

The maximum wear rate was determined manually. Therefore, each value could be analyzed, avoiding outliers such as punctual peaks which could be identified as artifacts. The color code, given in a 3D comparison mode, was modified until the maximum loss was indicated. Applying this scale mode, it was possible to check if the area of maximum wear was located in the corresponding area of different IOS and WLP. As six datasets of each wear status were available, the mean maximum vertical height loss and the standard deviation of each group (OC non-p, OC p, Tr3 non-p, Tr3 p, TD) was calculated with Excel 2016. Divergences from the reference measurements obtained from WLP were expressed in μm and as percentage deviations.

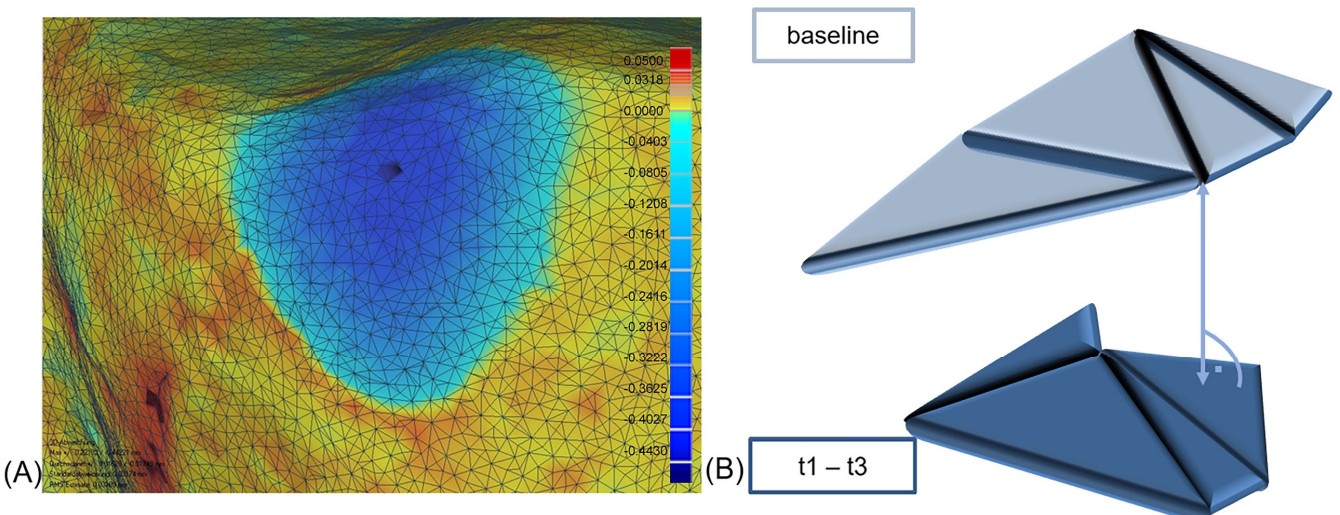

**Figure 3.** The principle of vertical height loss measurement is illustrated. The wear area of tooth 26 is presented in 3D comparison mode in Geomagic Qualify (**A**). Part (**B**) displays more detailed the program's method of determination of vertical deviations by dropping a perpendicular.

### 2.3. Evaluation of the Accuracy of Each Intraoral Scanning System (IOS)

The accuracy combines two components: trueness and precision (DIN ISO 5725-1) [15]. Trueness investigates the deviations from the test dataset to the reference dataset. In contrast, precision means the intergroup's deviation analysis and indicates repeatability. In order to determine the trueness, each of the six scan datasets of OC non-p, OC p, Tr3 non-p, Tr3 p and TD at t0–t3 were superimposed and compared with the respective reference dataset, WLP; (n = 6). To acquire precision values, the scan datasets of the IOS at t0–t3 were compared to each other (each scanner; including the modifications: p and non-p per wear stage: n = 15). Incorporating the whole tooth surfaces 24–28, the assessment of accuracy was conducted with Geomagic Qualify 2012 performing the best-fit-alignment and 3D comparison mode.

### 2.4. Using Cross Sections for 2D Analysis

To examine whether specific surface textures in the different datasets of WLP and of the IOS led to some outliers, with respect to wear measurement, the different data acquisition systems' surface profile was investigated. Thus, the 2D analysis visualized the different modes of surface representation of an intraoral scanning system compared to WLP. For the wear stages t1, t2, and t3, one dataset each out of the six captured by OC, Tr3, and TD was randomly selected. Since there was only one dataset from WLP available, this one was used. All datasets were superimposed in order to align them within the same coordinate system. This procedure was mandatory to define an identical cross-section for all datasets in the area of maximum vertical height loss. In Figure 4 for WLP, Figure 5 for TD, Figure 6 for Tr3, and Figure 7 for OC, the yz-planes are shown as they were identically placed as described above. The sections display the wear status at t1, t2, and t3 in relation to the baseline dataset t0, which is represented by the purple line. The cross-sections were evaluated qualitatively based on Figures 4–7:

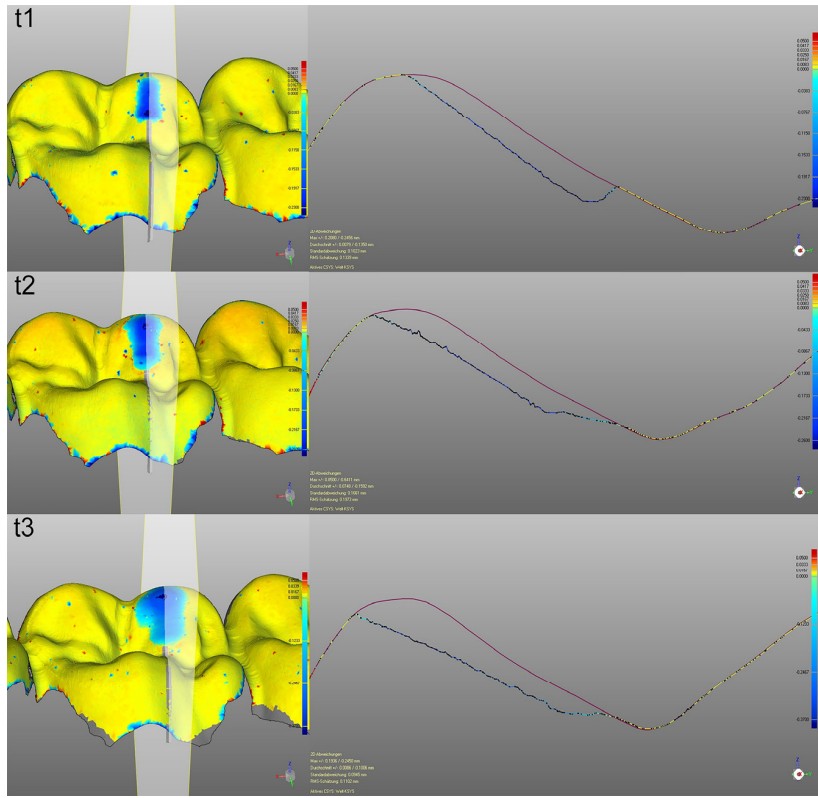

**Figure 4.** The position of the plane for 2D analysis at the three stages of wear acquired by WLP is shown on the left. Besides, the cross-section of each wear area is presented on the right.

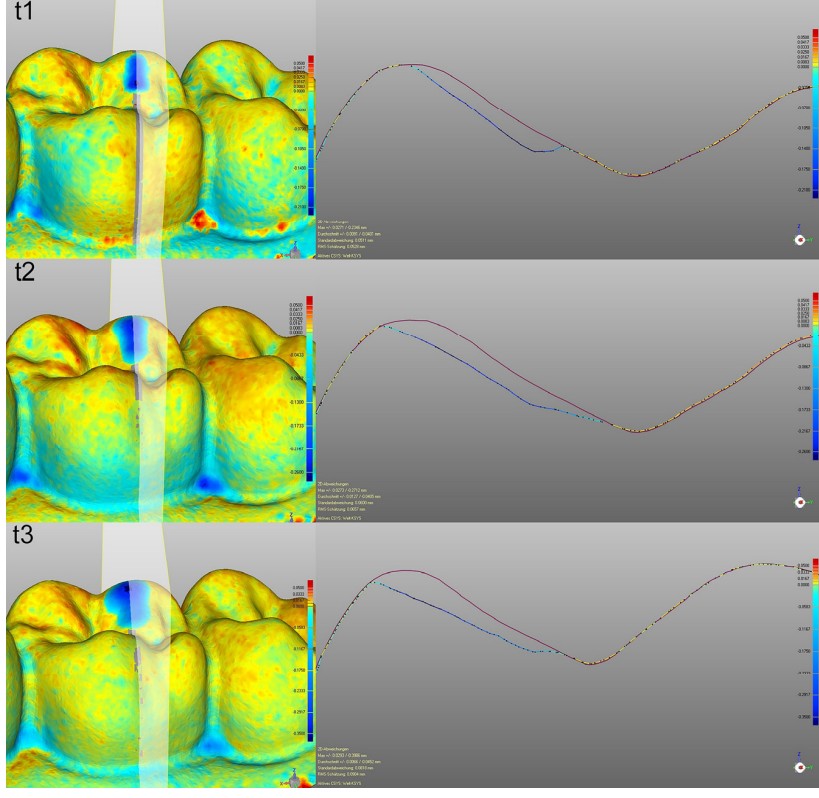

**Figure 5.** The position of the plane for 2D analysis at the three stages of wear acquired by TD is shown on the left. Besides, the cross-section of each wear area is presented on the right.

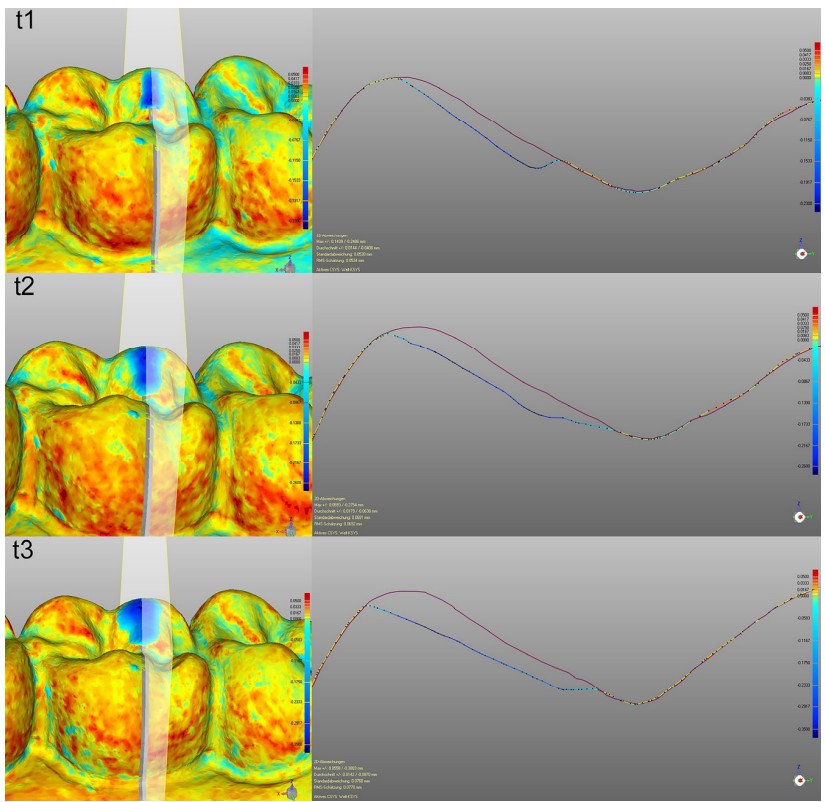

**Figure 6.** The position of the plane for 2D analysis at the three stages of wear acquired by Tr3 is shown on the left. Besides, the cross-section of each wear area is presented on the right.

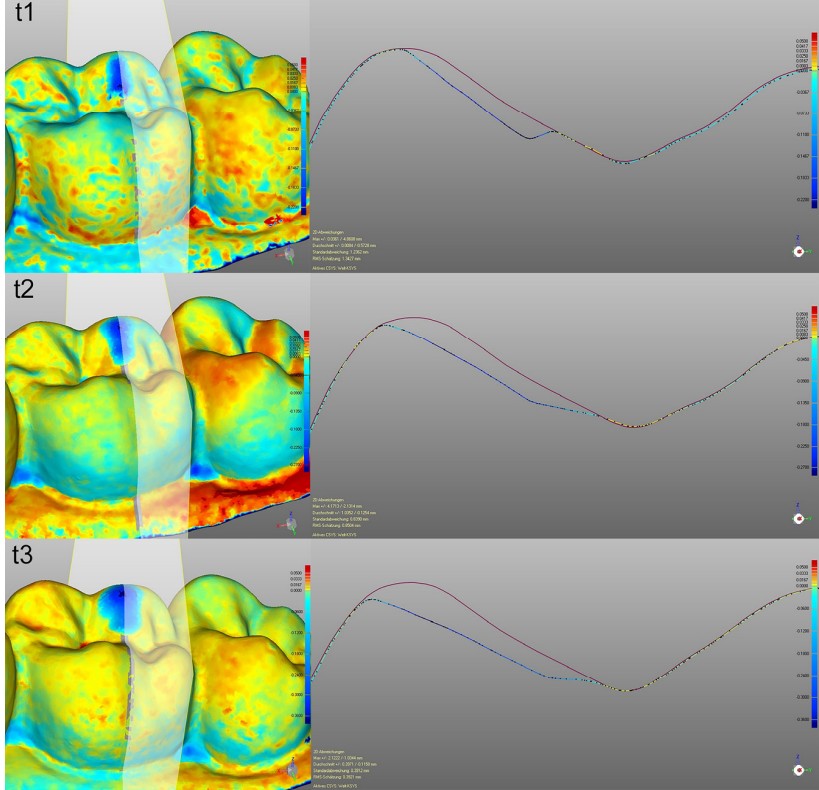

**Figure 7.** The position of the plane for 2D analysis at the three stages of wear acquired by OC is shown on the left. Besides, the cross-section of each wear area is presented on the right.

*2.5. Statistical Analysis*

The program G*Power 3.1.9.4 [16] was applied for a post hoc power analysis (α err prob = 0.05, Power (1-β err prob) = 1.0, effect size f = 2.011, and the total sample size was 36). The data were evaluated descriptively with Excel 2016. The Kolgomorov–Smirnov test was performed to evaluate the normal distribution of the results of wear measurement and accuracy evaluation. In order to test for statistically significant differences (α = 0.05) of the measured wear values, the Wilcoxon signed-rank test (Bonferroni corrected) was applied. The Kruskal–Wallis test was used to analyze statistically significant differences within the different groups of IOS in the context of accuracy measurements.

## 3. Results

The Kolgomorov-Smirnov-test revealed a normal distribution for all groups since the significance was greater than α > 0.05.

*3.1. Wear Measurement on the Basis of the IOS Compared to WLP*

3.1.1. Wear Values at the Wear Step t1

As can be seen in Table 1, at t1 a mean maximum vertical height loss between 229 μm and 237 μm was obtained for tooth 27 on the basis of the different IOS. For tooth 26, the mean maximum wear results ranged from 217 μm to 239 μm, when using IOS for data acquisition. Using WLP, wear of 227 μm was measured at FDI 26 and 221 μm at FDI 27. Tr3 p had the largest deviation from WLP with +7%.

**Table 1.** Wear values at the wear step t1.

|  | **FDI 26 (t1)** | **FDI 27 (t1)** |
| --- | --- | --- |
| WLP (reference) | 227 | 221 |
| OC non-p | 217 (±4) + 4% | 232 (±4) + 5% |
| OC p | 239 (±7) + 5% | 234 (±6) + 6% |
| Tr3 non-p | 231 (±2) + 2% | 235 (±6) + 6% |
| Tr3 p | 223 (±13) − 2% | 237 (±6) + 7% |
| TD | 220 (±2) − 3% | 229 (±6) + 4% |

3.1.2. Wear Values at the Wear Step t2

Applying IOS at t2, wear data at tooth 27 ranged from 267 μm to 287 μm. Wear measured on the basis of WLP was 242 μm. At tooth 26, values based on the IOS were between 314 μm and 330 μm, and WLP revealed 300 μm. The percentage deviation exceeds 10% in 4 out of 10 cases up to 19% at tooth 27, like it is presented in Table 2.

**Table 2.** Wear values at the wear step t2.

|  | **FDI 26 (t2)** | **FDI 27 (t2)** |
| --- | --- | --- |
| WLP (reference) | 300 | 242 |
| OC non-p | 328 (±9) + 9% | 267 (±7) + 10% |
| OC p | 329 (±11) + 10% | 287 (±11) + 19% |
| Tr3 non-p | 330(±9) 10% | 276(±13) + 15% |
| Tr3 p | 320 (±9) + 7% | 281 (±8) + 16% |
| TD | 314 (±7) + 5% | 273 (±9) + 13% |

3.1.3. Wear Values at the Wear Step t3

The mean maximum wear values for the last simulated wear level are shown in Table 3. Whereby, the highest wear value obtained from the IOS was 390 μm at tooth 27. The WLP data revealed 344 μm. At t3, there was a range of 415 μm to 441 μm for the IOS at tooth 26. The WLP measured 417 μm. At this point, 3 out of 10 cases showed a maximum deviation of +13%.

**Table 3.** Wear values at the wear step t3.

|  | **FDI 26 (t3)** | **FDI 27 (t3)** |
|---|---|---|
| WLP (reference) | 417 | 344 |
| OC non-p | 415 (±6) + 0% | 371 (±4) + 8% |
| OC p | 428 (±10) + 3% | 387 (±7) + 13% |
| Tr3 non-p | 441 (±6) + 6% | 361 (±9) + 5% |
| Tr3 p | 434 (±10) + 4% | 390 (±12) + 13% |
| TD | 432 (±4) + 5% | 328 (±9) + 11% |

The comparisons of the wear values within the IOS did not show statistically significant differences between the different intraoral optical impression systems ($\alpha < 0.05$, Bonferroni corrected).

### 3.2. Accuracy Values of the IOS

The results of the accuracy evaluation and the distribution of those are shown in Table 4 and Figure 8. Trueness values ranged from 22.5 μm (±4.8 μm) OC non-p to 12.8 μm (±0.7 μm) Tr3 non-p, whereby the results for precision varied between 19.5 (±4.5) μm OC non-, Tr non-p and 10.7 μm (±5.2 μm) OC p.

**Table 4.** Accuracy (trueness and precision) values for the applied IOS.

|  | **IOS** | **Trueness [n = 6]** **Mean (±SD) [μm]** | **Precision [n = 15]** **Mean (±SD) [μm]** |
|---|---|---|---|
|  | OC non-p | 22.5 (±4.8) | 19.5 (±4.5) |
|  | OC p | 14.6 (±0.9) | 13.6 (±7.6) |
| t0 | Tr3 non-p | 14.9 (±1.8) | 19.5 (±7.1) |
|  | Tr3 p | 15.7 (±1.2) | 15.1 (±3.7) |
|  | TD | 17.1 (±1.0) | 11.9 (±2.8) |
|  | OC non-p | 22.5 (±1.4) | 12.1 (±3.2) |
|  | OC p | 14.3 (±0.4) | 15.3 (±7.5) |
| t1 | Tr3 non-p | 12.8 (±0.7) | 18.0 (±5.3) |
|  | Tr3 p | 15.7 (±0.6) | 17.2 (±7.0) |
|  | TD | 20.6 (±1.4) | 17.5 (±4.2) |
|  | OC non-p | 21.8 (±1.7) | 14.7 (±3.8) |
|  | OC p | 17.6 (±0.8) | 11.2 (±3.5) |
| t2 | Tr3 non-p | 14.5 (±1.2) | 14.6 (±4.5) |
|  | Tr3 p | 14.2 (±1.3) | 13.9 (±5.4) |
|  | TD | 19.9 (±1.4) | 15.3 (±5.0) |
|  | OC non-p | 19.2 (±1.4) | 16.7 (±4.2) |
|  | OC p | 21.8 (±1.3) | 10.7 (±5.2) |
| t3 | Tr3 non-p | 13.6 (±1.3) | 14.5 (±4.3) |
|  | Tr3 p | 18.3 (±2.2) | 12.2 (±5.2) |
|  | TD | 21.3 (±1.2) | 11.6 (±3.3) |

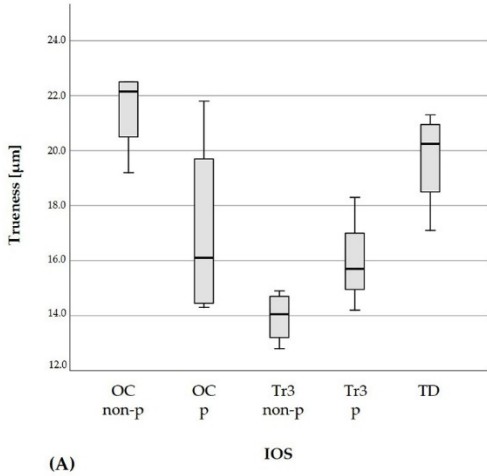
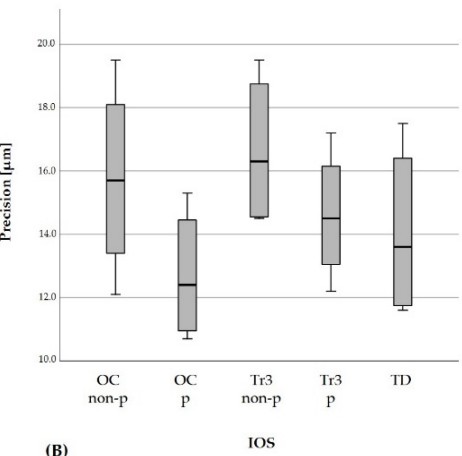

**Figure 8.** The distribution of the accuracy values (Trueness (**A**)and Precision (**B**)) based on the Kruskal-Wallis-test is shown by the boxplot diagrams. The box shows the range of the calculated values and the median value.

Tr3 non-p showed statistically significant lower trueness results than TD and OC non-p, as well as the powdered variation of Tr3 in comparison to OC non-p.

The statistical results concerning the precision of the different IOS did not differ significantly.

### 3.3. Using Cross Sections for 2D Analysis

The surface profile of the IOS and WLP is shown in Figures 4–7. The IOS present the worn areas as even regions, as can be seen in Figures 5–7. WLP in Figure 4, exhibits another perspective on the surface quality of the wear areas. At t1 and t3, an evenly wavy surface is shown, whereas, at t2, large peaks are visible. Comparing the two imaging systems, it becomes evident that the IOS do did not provide the same level of detail as WLP.

## 4. Discussion

### 4.1. Material and Method

Due to its thermal and optical properties, zirconia is a suitable material for in-vitro accuracy measurements. Therefore, the zirconia was the material of choice in the present study [17–19]. In terms of the concept of the ground truth, a reference model is established that can be used to collect datasets that serve as a reference (gold standard) to study the performance of the acquisition systems under investigation like it is described by Cardoso et al. [20]. Furthermore, various research groups investigating the accuracy of optical impression systems follow this workflow: one master model was digitized with both a high-accuracy optical impression system to establish a reference dataset and the optical impression system under investigation [7,8,21–23]. Commercially available intraoral optical impression systems have the potential to simplify clinical wear measurement. However, there is a need to investigate their suitability compared to conventional methods [13].

According to the consensus paper "Severe Tooth Wear: European Consensus Statement on Management Guidelines" [24], it is of major interest whether tooth wear shows physiological or pathological patterns. Based on this decision, the management of tooth wear is characterized by multi-layered aspects of diagnosis, patient enlightenment, and monitoring to clarify the severity of wear, wear progress, and its probably multifactorial origin. Especially TWES [25] is a well-established and further developed (TWES 2.0) tool in the context of the multifaceted complex of tooth wear diagnosis and management. Nevertheless, Wetselaar et al. define the use of intraoral scanning systems as a prospective standard in the dentist's daily routine and as a supporting tool alongside taxonomies like TWES. Additionally, it appears advantageous to explore different approaches and combine multiple benefits of different systems to provide the best possible treatment for patients.

Advantages of intraoral scans are their ease of production, the storage of the 3D data for permanent documentation and availability, and the possibility of quick analysis of the entire arch by superimposing different scans. By true color display, the character of the wear facets is also visible. Consequently, the detection, analysis, treatment, and monitoring of tooth wear require various features to arrive at therapeutic decisions [26].

The key prerequisite for the introduction of digital wear measurement on the basis of digital intraoral impressions as a standard method is the sufficient accuracy of the digital workflow. Therefore, the optical impression systems were compared with the optical profilometry, which is the gold standard and shows the highest accuracy in wear measurement with a resolution of 0.2 μm in vertical direction [2,27,28]. The test was performed as a direct, pure comparison of the acquisition systems because the IOS data were compared with the data from WLP, obtained by direct profilometry of the samples. In clinical reality, using WLP as a measurement tool, several additional steps are necessary, like impression taking and cast production. These steps were avoided intentionally in order to obtain the pure profilometry data without any bias. In contrast, to the investigation of Hartkamp et al., the novelty of the present study is that the study model is made out of zirconia, due to the fact that metallic surfaces lead to higher accuracy values [29]. Lava C.O.S. was very accurate but a mandatory powder system. Therefore, this investigation considered powder-free scanning systems to check if they can provide similar results.

### 4.2. Method of Wear Simulation and Data Acquisition

Six datasets per IOS were performed. The power estimation revealed that the number of data sets was sufficient, which can be explained by the low standard deviations. WLP as a reference method was performed once. At t1, the simulated wear was 230 μm. Compared with in-vivo wear results, this is a relatively high value [2,3,27,30]. In further investigations, the wear simulations should be imitated by smaller steps ideally by a CNC-driven process [31–33]. Apart from the tooth and material-related wear, there is also the possibility of grinding procedures carried out by the dentist. Thus, a relevant side effect of the present study was whether intraoral scanners are able to capture artificial traces caused by grinding.

### 4.3. Discussion of the Results

The wear values investigated on the basis of intraoral scanners showed higher values than those obtained from the optical profilometry (Tables 1–3). 23 out of 30 measurements, based on the IOS, exhibited wear deviations of less than 10% compared to WLP. In order to decide from which values IOS are suitable for wear measurement, it was necessary to define a threshold value. Hartkamp et al. measured a maximum difference of 12.6% between the applied IOS and the optical profilometry [7]. Based on these findings, a threshold of 10% was established. The maximum deviation was 19% (+45 μm). One potential explanation for the outlier values (>10% from WLP) is that for the manual wear simulation a diamond bur was used. The cross-sections visualize that WLP detected the traces caused by the diamond bur when simulating the wear facets (Figure 4) by grinding. The cross-sections of the surfaces delivered by TD (Figure 5), Tr3 (Figure 6), and OC (Figure 7), show a smoother texture of the wear areas. The IOS were not able to detect the peaks in detail resulting from the grinding procedure. In particular, the pronounced peaks at t2 compared to t1 and t3 provide a rationale for explaining the increased deviations at the wear level t2 (tooth 27).

With respect to the present study, wear measurements based on the IOS led to higher wear results than those evaluated with the WLP data at the same stage because pronounced surface structures were captured less detailed by the IOS. Conventional wear measurement requires several working steps, such as impression taking and cast production. The digital workflow of the wear measurement is decisively determined by the data acquisition and the analysis strategies. Key aspects are the characteristic of the triangle meshes and the alignment and measuring strategies. Figure 9 shows the variety of triangular structures.

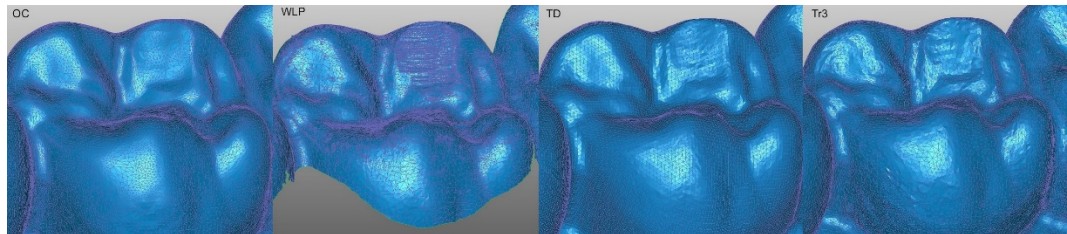

**Figure 9.** The variety of the triangular meshes of the different acquisition systems is shown.

Poor mesh quality is characterized by triangles with a low edge ratio or tiny interior angles and non-equilateral triangles. Extremely tilted triangles may be the cause of misleading wear measurements. High-quality meshes are defined by homogenous triangles, comprising the same homogenous, equilateral triangles in their neighborhood while keeping the detail and edge sharpness of the geometry [34,35]. In future investigations, it has to be elaborated to what extent the mesh quality and a kind of re-meshing is necessary for the context of wear measurement. The three most common strategies for superimposing datasets are an exclusively landmark-based approach, a standard best fit alignment, and a reference best fit [12,36,37]. In terms of wear measurement, the exclusively landmark-based alignment is inaccurate, especially due to the difficulty of positioning landmarks with micrometer accuracy. In contrast, the standard best-fit alignment, including the whole surface of the datasets, is based on an iterative closest point (ICP) algorithm. The absolute distance between two datasets is minimized as far as possible. Another modification of the standard best fit alignment is the reference best fit. If it is possible to determine which areas of the scanned tooth surface have been worn, these changes can be excluded. The operator manually determines the areas to be included in the alignment. Although the reference best fit is user-dependent and therefore more error-prone, O'Toole et al. were able to achieve the most accurate results in their investigation. Applying the exclusive landmark-based alignment or the standard best fit alignment, wear has been underestimated [12]. Therefore, the reference best-fit alignment is the preferred alignment strategy. Due to the manual wear simulation, the worn areas were known and excluded to achieve the best possible alignment. In clinical situations, this is only possible to a limited extent by identifying grinding and clenching facets. To avoid the consideration of wear facets during best fit calculation a threshold value should be defined so that the algorithms exclude surfaces from the registration process if they differ too much.

If excluding the worn regions, the remaining areas of the scans should fit perfectly. However, every scan differs depending on the accuracy (trueness and precision) of the scanning system. Zimmermann et al. measured local accuracy values. The term local referred to analyses that were limited to a single tooth preparation. The authors found trueness values of 36.7 µm for OC and 23.3 µm for Tr3. In terms of precision, 14.0 µm Tr3 and 20.3 µm OC were reported [21]. According to Ender et al., Tr3 showed a precision of 15.5 µm ± 1.7 µm and trueness of 27.5 µm ± 1.8 µm when capturing a posterior tooth segment (13–17). OC exhibited a precision of 18.8 µm ± 4.1 µm and trueness of 28.9 µm ± 3.2 µm [22]. The mean trueness of a single-tooth acquisition by Lee et al. was 13.8 ± 1.4 µm (max. 16.3 µm, min. 12.0 µm) when OC was used. The precision was 12.5 ± 1.4 µm (max. 18.8 µm, min. 7.7 µm) [23].

Deviations of trueness and precision depend on the applied scanning system and the surface material to be scanned. Dutton et al. examined the impact of different materials on the scanner's accuracy. Scanning a zirconia crown, OC exhibited a mean trueness of 29.3 µm ± 5.7 µm and Tr3 22.1 µm ± 5.7 µm. Whereas Tr3 showed no statistically significant influence on the accuracy due to the surface material. IOS applying the optical principle of active triangulation, like OC are more prone to errors [38]. However, Lim et al. compared metallic and non-metallic materials. Concerning the applied IOS, Tr3 revealed significantly worse results concerning the metallic materials. Additionally, the impact of surface conditioning (application of powder) was focused on. Comparable to the findings

of the present study, no significant differences were identified [29]. It has to be emphasized as a limitation of this investigation that the present results refer to scanned zirconia surfaces, but no additional material beyond that.

The literature reports an average accuracy of about $+/-20$ µm in relation to a single tooth measurement, which correlates with the present accuracy values of this study. In particular, OC non-p exhibited the highest values for trueness (22.5 µm–19.2 µm) and the highest deviation in wear measurement (45 µm) at the same time. Therefore, a mean maximum error of approximately $+/-40$ µm could be postulated if two digital impressions of different wear statuses are compared with each other. This corresponds to the maximum deviation of 45 µm of the IOS compared to WLP.

## 5. Conclusions

The null hypothesis can be confirmed to a limited extent. The investigated digital impression systems are a promising alternative to wear measurement based on WLP due to their simple application, although inaccuracies of $+/-20$ µm per partial arch impression have to be taken into account. Limitations of this study included the examination of only one material (zirconia), the scanner's disability to capture needle-like structures, and the extended simulated wear. These shortcomings have to be weighed against the ability to capture surface changes in a reasonable, quick, and user-friendly way so that high volumes of 3D data can be processed nearly in real-time for patient monitoring and scientific purposes.

**Author Contributions:** Conceptualization, C.K. and S.R. (Sven Reich); Data curation, C.K., U.L. and S.R. (Sven Reich); Investigation, C.K., S.R. (Stefan Raith) and S.R. (Sven Reich); Methodology, U.L., S.R. (Stefan Raith) and S.R. (Sven Reich); Project administration, C.K., U.L., S.R. (Stefan Raith) and S.R. (Sven Reich); Supervision, S.R. (Stefan Raith) and S.R. (Sven Reich); Visualization, U.L., S.R. (Stefan Raith) and S.R. (Sven Reich); Writing—original draft, C.K. and S.R. (Sven Reich); Writing—review & editing, C.K., U.L., S.R. (Stefan Raith) and S.R. (Sven Reich). All authors have read and agreed to the published version of the manuscript.

**Funding:** This research received no external funding.

**Institutional Review Board Statement:** Not applicable.

**Informed Consent Statement:** Not applicable.

**Data Availability Statement:** The data presented in this study are available on request from the corresponding author.

**Acknowledgments:** The study was supported by Vita Zahnfabrik providing Vita YZ HT ceramic and 3Shape providing the Trios3 scanner.

**Conflicts of Interest:** The author Sv.R. has or had commercial relationships with the following companies/foundations with respect to consulting and presentation activities and third-party-funded research: 3M OralCare, 3Shape, Amann Girrbach, Camlog, Oral Reconstruction Foundation, DCS, Dentaurum, Dentsply Sirona, Ivoclar Vivadent, Straumann, Vita Zahnfabrik. The authors Chr.K., St.R., and Ul. L. declare no conflict of interest.

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
