# Peer review of "Measurement of Tooth Wear by Means of Digital Impressions: An In-Vitro Evaluation of Three Intraoral Scanning Systems"

_applsci, doi:10.3390/app11115161_

Round 1

Reviewer 1 Report

This article is aimed to investigate if the intraoral scanner is comparable to a high-resolution non-contact white light profilometer in terms of measuring tooth wear. The topic is lack of clinic interest and there are some limitations in the methods.

There are varied ways to evaluate tooth wear. The most popular approach is using Tooth Wear Evaluation System by Wetselaar et al, which evaluated by the eyes of a dentist and no special tool is needed. Moreover, it can be evaluated by the wear surface area, wear depth, wear surface roughness. The current study only compared the wear depth. For the roughness, the study only provided some subjective description

Line 45 "The scanner's optical principle was based on wavefront sampling, and "dusting" of the scanned surface with titanium dioxide was required." This statement is incorrect. Different intraoral scanners have varied optical principles. Both Trios and CEREC AC are using Confocal Microscopy and Active triangulation. A dusting of the object is not required for Trios and CEREC systems.

Line 51. Why choose "measurement within deviations of a maximum of 10%." as the principal method to decide if the intraoral scanner is comparable to a white light profilometer?

I would suggest the author using Trueness and Precision which are more universal evaluating methods.

I would suggest using

Figure 2. It's hard to see the wear areas in figure 2. Please modify the picture to make it more obvious. You can zoom in, or add markers indicating the location of wear areas.

Author Response

Dear Reviewer,

Thank you! 

Reviewer 2 Report

Measurement of tooth wear by means of digital impressions:

An in-vitro evaluation of three intraoral scanning systems

Summary of manuscript: This study aimed to test if wear measurements based on data files obtained with powder-free scanners are comparable to results based on profilometry. This report concludes that the investigated digital impression systems are a promising alternative to wear measurement based on due to their simple application. First of all, I would like to express appreciation for your effort. However, the reviewer feels that it is not enough to accept it at the present form and several issues that needs to be addressed.

General comment

  • In this in vitro experiment, was only one zirconia cast prepared? In order to validate the universality of the results, it is necessary to consider using multiple models in same shape.
  • In these experiment, it is necessary to consider the accuracy of each scanner itself. Moreover, have author considered the effect of the range of superposition?
  • The novelty of this manuscript, compared to the past study, should be described.
  • The result of normality in this experimental data and clinical implication should be added.

Materials and methods

  • Author mentioned the α value regarding the sample size calculation, however it’s not enough. Author should clearly mentioned α, β value, effect size and appropriate sample size.

Conclusions

  • The conclusions of the abstract and the manuscript are different.

Author Response

Dear Reviewer,

Thank you! 

Reviewer 3 Report

  1. What SEM abbreviation means? (line 35 and 39). Internet gives transcription like Search Engine Marketing
  2. When repeated intraoral scanning is done, the accuracy of the scan may be affected by the time intervals between repeated use of the scanner. It is one thing when the scanner is turned on and six repeated scans are made at once, one after the other, and another thing when the scanner is turned on six times on different days. When assessing the degree of tooth wear in real conditions, the interval between scans can be 0.5-1 year or more. What were the time intervals between repeated scans in your investigation?

Author Response

Dear Reviewer,

Thank you! 

Round 2

Reviewer 1 Report

The authors have addressed my concerns and have made proper revisions to the manuscript.

Reviewer 2 Report

Measurement of tooth wear by means of digital impressions:

An in-vitro evaluation of three intraoral scanning systems

Summary of manuscript: This study aimed to test if wear measurements based on data files obtained with powder-free scanners are comparable to results based on profilometry. This report concludes that the investigated digital impression systems are a promising alternative to wear measurement based on due to their simple application to capture surface changes in a reasonable and quick way.

This manuscript was well revised.